Corrected: Publisher correction

# Shape and fluctuations of frustrated self-assembled nano ribbons

Mingming Zhang[1,2], Doron Grossman [1], Dganit Danino[2] & Eran Sharon[1]

Self-assembly is an important process by which nontrivial structures are formed on the sub-micron scales. Such processes are governed by chemical and physical principles that dictate how the molecular interactions affect the supramolecular geometry. Currently there is no general framework that links between molecular properties and the supramolecular morphology with its size parameters. Here we introduce a new paradigm for the description and analysis of supramolecular structures that self-assemble via short-range interactions. Analysis of molecular interactions determines inputs to the theory of incompatible elasticity, which provides analytic expressions for supramolecular shape and fluctuations. We derive quantitative predictions for specific amphiphiles that self-assembled into chiral nanoribbons. These are quantitatively confirmed experimentally, revealing unique shape evolution, unusual mechanics and statistics, proving that the assemblies are geometrically incompatible. The success in predicting equilibrium and statistics suggests the approach as a new framework for quantitative study of a large variety of self-assembled nanostructures.

---

[1] The Racah institute of Physics, The Hebrew University of Jerusalem, Jerusalem, Israel. [2] CryoEM Laboratory of Soft Matter, Faculty of Biotechnology and Food Engineering, Technion—Israel Institute of Technology, Haifa, Israel. Correspondence and requests for materials should be addressed to E.S. (email: erans@mail.huji.ac.il)

A prototypical, interesting class of molecular assemblies is that of twisted and helical nanoribbons, assembled from chiral molecules. Such structures are formed by a wide variety of building blocks, such as amphiphilic lipids, peptide amphiphiles, amino acid derivatives, and proteins[1,2], e.g. during the evolution of some neurodegenerative disorders[3]. Many of these systems undergo morphological evolution during assembly: at early stages they form long, narrow twisted ribbons with a straight centerline, around which the ribbon twists with pitch $P$ (Fig. 1a). As assembly proceeds, the ribbons width, $W$, increases, leading to the growth of $P$. Further widening induces shape transition into helical ribbons whose centerline is a helix, characterized by its pitch $P$ and radius $R$ (Fig. 1b). Finally, as widening further proceeds, the helical ribbons close into tubes with radius $R$ (Fig. 1c). Currently, this common shape evolution is not understood and cannot be linked to the molecular chemistry and interactions. Moreover, it is not even known which kind of molecules it can be applied to.

The modeling of chiral ribbons is based on two main approaches[4,5]. The first analyzes the basic chemical interactions between adjacent molecules, in order to determine the relative bends and twists[6]. It is further assumed that the intrinsic geometry that is prescribed by the local interactions is accurately manifested by the supramolecular morphology. However, in many cases, in order to form suprastructures, the optimal nearest neighbors' configuration (the intrinsic geometry) must be distorted, since the relaxed elements would not fit together to form a continues aggregate. Such systems are known as geometrically frustrated[7]. The flexibility of most biomolecules and their soft interactions allow such systems to overcome the geometrical frustration via distortion with respect to the locally optimal configuration. This elastic distortion generates internal stresses, affecting the aggregation process[7], the shape[8–11] and the mechanical properties[12,13]. Therefore, in order to describe the global shape, a mechanical model is needed, in addition to characterization of the intrinsic geometry. This is the second approach, based on continuum mechanics, which models twisted and helical ribbons in various ways. These include liquid membrane models[14], as well as solid ribbons with broken[15,16], or unbroken[17] mirror symmetry. Such models are phenomenological and qualitative—not related to the specific chemical interactions. These limitations prevent quantitative comparison between experiments[18–21] and theory.

Recently, we used the theory of incompatible elasticity[22] to derive a two-dimensional (2D) modeling for shape selection of thin, geometrically frustrated, sheets[23]. The intrinsic geometry serves as input to the theory, and can originate from different processes, e.g., plastic deformations[24], active swelling[25,26] or growth of biological tissues[27]. The theory was successfully applied to analyze equilibrium shapes of macroscopic ribbons having intrinsic twist[28]. Additionally, a reduced, one-dimensional (1D) model of incompatible ribbons[29], provides analytical expressions for equilibrium shapes, as well as for statistical properties of thermal ribbons. Considering self-assembled nanoribbons, one can use the intrinsic geometry, computed from the chemical interactions, as described in Nandi and Bagchi[6] (the first approach) as input to the 1D elasticity theory, in order to analytically compute ribbon shape and fluctuations. This new combined methodology integrates the two approaches, proposing a new paradigm e.g., for modeling self-assembled nanoribbons, with an unprecedented quantitative link between molecular and supramolecular properties. It is applicable to a wide variety of self-assembled slender structures (see Supplementary note (5))

Here we perform extensive cryo-electron microscopy (cryo-TEM) shape measurements of an amphiphile, $C_{12}$-$\beta_{12}$ (N-α-lauryl-lysyl-aminolauryl-lysyl-amide), as it assembled into twisted nanoribbons and further, to helical ribbons and tubes (Fig. 1). We write the 1D elastic model for the ribbons and provide analytical expressions that describe the ribbon's equilibrium shape over the entire range of width. The geometrical parameters in the model are determined by the interaction between monomers. We go beyond studying equilibrium configurations by analyzing shape fluctuations of ribbons. We measure the predicted unusual statistics, which indicate softening of the ribbon with increasing width. These results show that the self-assembled ribbons are indeed frustrated ribbons, well captured by the combined chemical-physical approach. Finally, we discuss how the approach can be applied to other self-assembled ribbons.

## Results

**Equilibrium configurations.** We start by computing ribbon's equilibrium configurations, using the theory of incompatible elastic sheets, with parameters determined by the molecular interactions. The theory uses two input fields that encode the intrinsic geometry. The reference metric $\bar{a}$ is determined by gradients in equilibrium distances within the plane of the ribbon. The reference curvature $\bar{b}$ is determined by gradients of equilibrium distances across the ribbon's thickness. In the gel phase (at 25 °C) $C_{12}$-$\beta_{12}$ (Supplementary Fig. 1) adopts a bolaamphiphile-like configuration (Fig. 2a) and forms ribbons reminiscent of a lipid bilayer[20], with hydrophilic heads facing out and hydrophobic carbon chains hidden inside the sheet (Fig. 2b). Neighboring headgroups interact via hydrogen bonds between amide

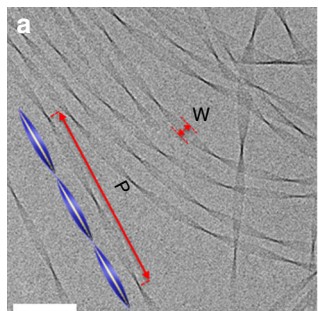
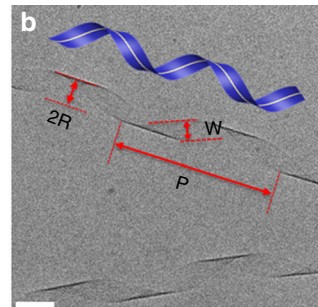
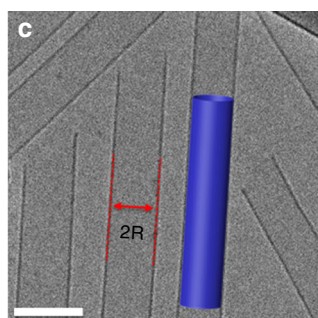

**Fig. 1** Shape evolution and characterization: Cryo-TEM images and illustrations (insets) of self-assembled N- α -lauryl-lysyl-aminolauryl-lysyl-amide ($C_{12}$-$\beta_{12}$) ribbons. **a** After ~24 h of assembly most ribbons are twisted, having a straight centerline, (inset, yellow dashed line) i.e. $R = 0$. **b** After 1 week, helical ribbons are abundant. Their center line is a helix with given pitch, $P$, and radius, $R$. Determination of $W$, $P$, and $R$ from the image is demonstrated. **c** After 5 months most assemblies are tubes (distinguished by the dark parallel boundaries compared to the pale ends) with diameters $D = 2R \approx 100$ nm. Scale bars = 100 nm

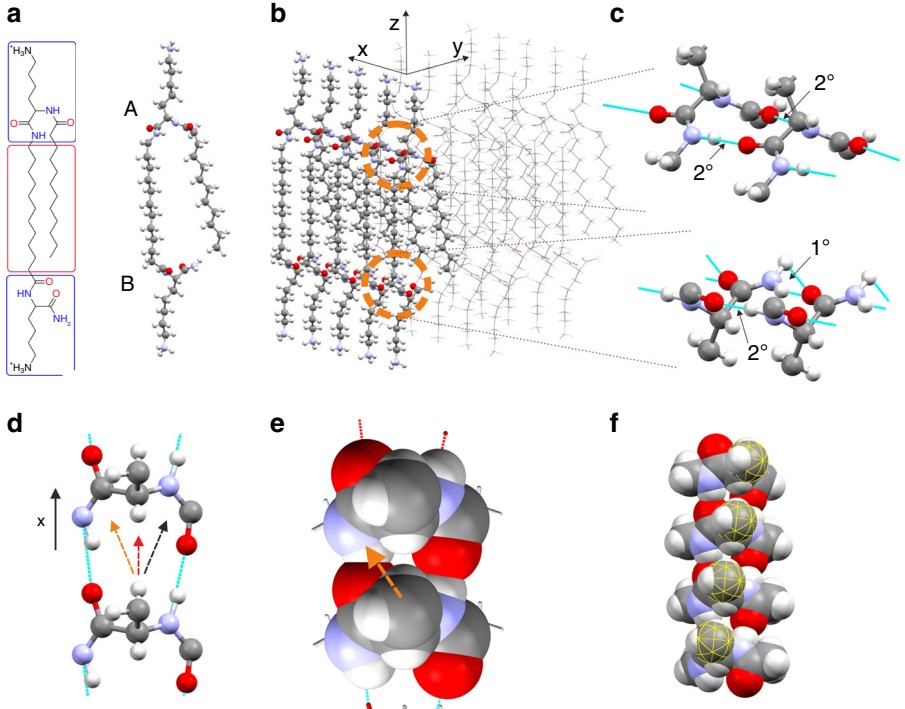

**Fig. 2** Dominant chemical interactions and the generation of intrinsic twist: **a** A Skeletal structural formula (left) and a ball-and-stick model (right) of $C_{12}$-$\beta_{12}$ in its bolaamphiphile-like configuration. The heads and chains are marked with blue and red respectively. At side $A$, the head contains two secondary amines, and at $B$, one primary and one secondary amine. **b** Hydrophobic interactions drive the assembly of $C_{12}$-$\beta_{12}$ into a ribbon, with hydrogen bonds forming along the x direction—the ribbons' long axis. The domains within the orange circles are magnified in **c**, illustrating the hydrogen bonds (cyan lines) on sides $A$ (secondary amines (2°)) and $B$ (one secondary and one primary (1°) amines). This asymmetry leads to asymmetry between the two faces of the ribbon (the lysine are truncated in panels **c**–**f**, for clarity). **d** A top view (in the x-y plane) of two ($S$) left-handed head groups (side $A$). Close packing of the two heads directs the methylene group into the largest free volume. Right (orange arrow) zero (red arrow) and left (black arrow) optional twists are illustrated. **e** The same headgroups illustrated with the VdW radiuses. The chirality (in this case the difference in VdW radiuses) leads to a larger free space on the left (orange arrow) than on the right. **f** The conformation in **e** induces right-handed intrinsic twist along the x direction

groups. This attractive interaction prescribes a linear order in the sheet, pulling neighboring heads tightly together. The optimal conformation between heads can be approximated by a close packing[6], inducing twist around the amide bonds with a preferred twist angle, $\theta_0 \sim 20^0 - 40^0$ between adjacent head-groups (when ignoring the effect of carbon tails) (Fig. 2e,f). Using here $S$ (left handed) chiral carbons, right-handed twist is preferred (see Fig. 2e and Supplementary Figs. 3–7). Next we consider the chain's Van der Waals (VdW) attractive interaction, between the 22 methylene groups. This interaction tends to align the molecules, resisting twist and is, therefore, minimal at $\theta = 0$. The combined energy associated with a given twist angle, $\theta$, between two amphiphils is, therefore, approximated by $E(\theta) \propto D^2(\theta - \theta_0)^2 + L^3\theta^2$, where the first/second term corresponds to the head/chain energy, respectively. Here, $D$ is an effective headgroup diameter and $L$ is the chain length. The optimal twist between two molecules is obtained by minimizing the energy with respect to $\theta$. We find the twist angle, $\theta^*$, to be in the range of $0.3°-2°$ (see supplementary note (2)), leading to spontaneous twist (angle per unit length) $k_0 = \frac{\theta^*}{D}$. Using $D = 0.6$ nm, we get $k_0 = 0.03 \pm 0.02 \frac{rad}{nm}$. The slight difference, $\Delta d$ in equilibrium length of the hydrogen bonds between primary and secondary amines (Fig. 2c) induces curvature, in addition to the twist. We mark it $\alpha k_0$, where $\alpha$ is a measure for the up-down asymmetry ($\alpha = 0$ implies a symmetric ribbon, as in refs. [15,18,28]). The curvature is approximately $\alpha k_0 \approx \frac{\Delta d}{Dt}$ and we find $\alpha = 0.1 \pm 0.07$. We note that a much more accurate estimations of $k_0$ and $\alpha$ can be achieved via molecular simulation. The reference

curvature tensor, which represents the right-handed twist, $k_0$, and the bend $\alpha k_0$ is:

$$\bar{b} = \begin{pmatrix} 0 & k_0 \\ k_0 & \alpha k_0 \end{pmatrix}$$

where we use a coordinate system aligned with the ribbon as in Fig. 2b (see also Supplementary Fig. 8). Finally we note that the ribbon has no structural lateral gradients hence, its reference metric is flat ($\bar{a}$ is the identity matrix). A similar approach can be implemented for the study of other molecular systems, where different molecular interactions and dimensions would determine different $\bar{b}$ and $\bar{a}$ (see examples in Supplementary note (5)).

Computation of ribbon equilibrium configurations consists of plugging $\bar{a}$ and $\bar{b}$ into the energy functional of the 1D theory[29] and solving the resulting Euler-Lagrange equations. The solution depends on the ribbon thickness, $t$, and width, $W$, as well as on its material properties. Here we limit the analysis to the case of isotropic elasticity, characterized by the Young's modulus, $Y$, and Poisson's ratio, $v$. The solutions are right-handed ribbons with radius $R$ and pitch $P$:

$$R(W) = \frac{l(W)}{l(W)^2 + m(W)^2} \qquad (1)$$

$$P(W) = \frac{2\pi m(W)}{l(W)^2 + m(W)^2} \qquad (2)$$

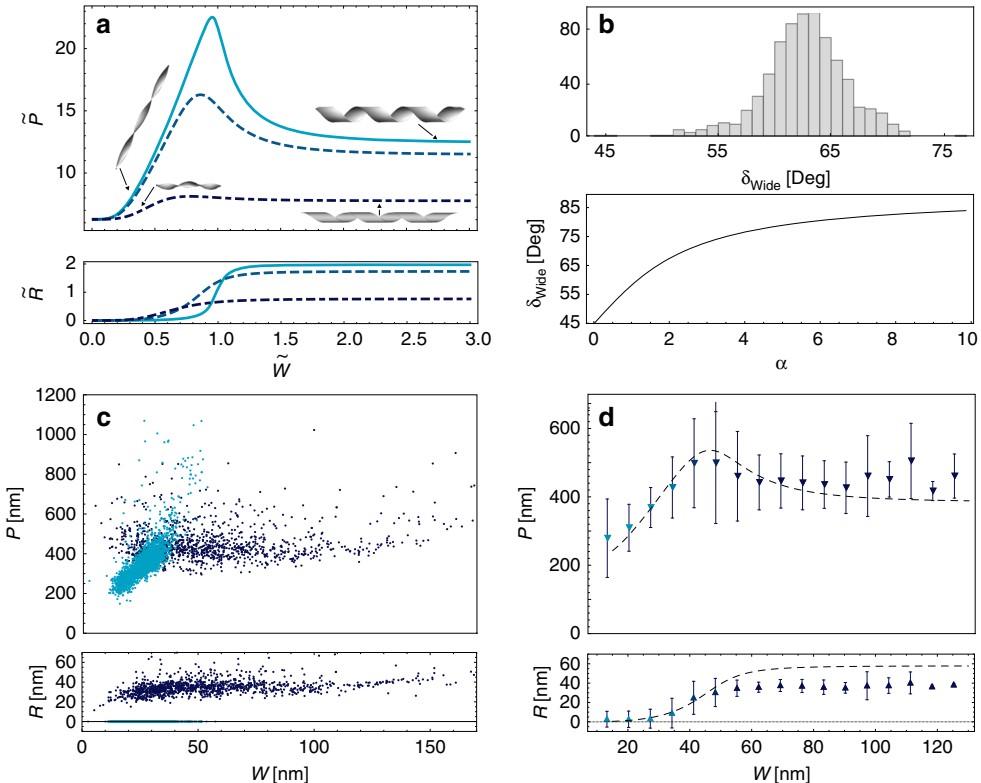

**Fig. 3** Ribbon's shape as a function of width. **a** Analytic solutions for the dimensionless ribbon-pitch, $\tilde{P}$, and radius, $\tilde{R}$, as functions of the dimensionless width, $\tilde{W}$ for different values of $\alpha$ (0.01 (solid cyan), 0.1 (dashed light blue), and 1 (dot-dashed dark blue)). As $\alpha$ increases, the twist-to-helical transition becomes smoother and occurs earlier. Insets show selected realizations of ribbon configurations. **b** Top: the distribution of the measured pitch angle of wide ($W > 80$ nm) ribbons. Bottom: the computed pitch angle of wide ribbons ($\tilde{W} \to \infty$),$\delta_{\mathrm{Wide}}$, as a function of $\alpha$. **c** Measurements of $P$ (top) and $R$ (bottom) vs. $W$. Cyan data points mark twisted ribbons ($R = 0$) and blue points indicate helical ribbons ($R > 0$). The pitch increases in the twisted phase, then slightly decreases, beyond the twisted-to-helical transition, and then both pitch and radius are stabilized on roughly constant values. Measurement error is typically ±6 nm for $P$ and ±4 nm for $R$. Note the large asymmetric scatter in the pitch near the twisted-to-helical transition at $W \approx 40$ nm. **d** The average (over $\Delta W = 2$ nm) of the data in **c** vs. $W$, together with the solutions for Eqs. (1) and (2) with $k_0 = 0.03 \frac{\mathrm{rad}}{\mathrm{nm}}$, $\alpha = 0.1$, $t = 3.4$ nm and the fitted Poisson's ratio $v = 0.5$. The experimental data is colored by the relative abundance of twisted (cyan) and helical (blue) ribbons at a given width. Error bars indicate s.d.

where $l(W; \alpha, k_0, \nu, t)$ and $m(W; \alpha, k_0, \nu, t)$ are functions of $W$, for a given set of $(\alpha, k_0, v, t)$ (see Supplementary note (3)). In the wide limit ($W^2 \gg \frac{t}{k_0}$), the solution becomes independent of $W$:

$$R_{\mathrm{Wide}} = \frac{\sqrt{4 + \alpha^2}(1 - \nu) - \alpha(1 + \nu)}{2((1 - \nu)^2 + \alpha^2\nu)k_0} \quad (3)$$

$$P_{\mathrm{Wide}} = \frac{4\pi}{(2(1 - \nu) + \nu\alpha(-\alpha + \sqrt{4 + \alpha^2}))k_0} \quad (4)$$

The full solution (Eqs. (1) and (2)) describes a twisted ribbon which, upon widening, becomes helical. It can be expressed in a dimensionless form,$\tilde{R}(\tilde{W})$, and $\tilde{P}(\tilde{W})$ (Fig. 3a and Supplementary Fig. 9). These analytical solutions qualitatively resemble the numerical results in Armon et al. [28], however, they include in addition the dependence on $\alpha$, the asymmetry of the sheet. We find that as $\alpha$ increases, the maximal pitch decreases, the twisted-to-helical transition occurs at smaller values of $\tilde{W}$, and over a wider range of $\tilde{W}$. Additionally, the pitch angle, $\delta \equiv \frac{P}{2\pi R}$ of wide ribbons (Fig. 3b) obeys the relation $\tan \delta_{\mathrm{Wide}} = \frac{1}{2}(\alpha + \sqrt{4 + \alpha^2})$, i.e. $\delta_{\mathrm{Wide}}$ increases with $\alpha$. Note that for symmetric bilayers, where $\alpha = 0$, $\delta_{\mathrm{Wide}} = 45°$ as in Armon et al. [28].

We now quantitatively compare model predictions with experimental measurements. We analyzed more than 500 cryo-TEM images, collecting over 4000 measurements of $W$, $P$ and $R$

for ribbons at different stages of assembly (Fig.1). The width and pitch of a given ribbon were found to be remarkably uniform, with a variation of ±3% over lengths of order few microns (Supplementary Fig. 2), justifying the definitions of ribbon-width and ribbon-pitch. Plotting $P$ and $R$ versus $W$ clearly reveals the twist-to-helical transition (Fig. 3c). For $W < 40$ nm most ribbons are twisted and their pitch increases with $W$. For $W > 40$ nm, most ribbons are helical, and beyond $W = 60$ nm twisted ones are hardly present. As $W$ increases beyond $W \approx 40$ nm the pitch stops its increase, and gradually decreases before stabilizing on a width-independent value of $P \sim 400$ nm. This evolution is consistent with our model predictions (Eq. (4), Fig. 3a). Further, we measure $\delta_{\mathrm{Wide}} = 62° \pm 3°$ (Fig. 3b inset), indicating that indeed $\alpha > 0$. All ribbons were right-handed (Supplementary Fig. 7).

We plot the computed $P(W)$ and $R(W)$ using the parameters estimated from the molecular interactions, $k_0 = 0.03 \frac{\mathrm{rad}}{\mathrm{nm}}$, $\alpha = 0.1$ and ribbon thickness, $t = 3.4$ nm [20], with the Poisson ratio,$v = 0.5$, being the only fitting parameter (the effect of $v$ on ribbon's shape is presented in Supplementary Fig. 9). The theoretical curves provide a good description of the ribbon's shape over the entire range of widths (Fig. 3d), including the decrease in the pitch after the transition and its width-independent value at large $W$. This is the first successful analytical prediction of the entire shape evolution of the ribbons.

Interestingly, Fig. 3d shows a systematic deviation between the measured radius of wide ribbons and the theoretical predictions. Furthermore, though the average data (Fig. 3d) is well described by our model, the raw data of radius measurements (Fig. 3c bottom) suggest that the twisted-to-helical transition is 1st order, rather than 2nd order. Some of the deviations may result from simplifications in the model (e.g., rough estimation of geometrical parameters and the assumption of isotropic elasticity). However, a more important factor is that due to thermal fluctuations the ribbons are not in a mechanical equilibrium. We therefore turn to study the statistics of ribbon shapes.

**Thermal fluctuations**. A huge (typically >100 nm) scatter in the data of both pitch and radius is noted in Fig. 3c, much larger than our measurement accuracy, which is better than 6 nm. It results from fluctuations in the shape of the supramolecular structures around their energy minimum (Eqs. (1) and (2)). The probability of finding a ribbon of width $W$ in some configuration, whose energy is larger by $\Delta E$ from the energy minimum is $p(\Delta E) \sim e^{-\frac{\Delta E}{k_B T}}$ where $k_B$ is the Boltzmann coefficient and $T$ is the temperature. Unlike equilibrium shapes, fluctuations are directly related to the ratio between thermal, ($k_B T$), and elastic, ($\Delta E$), energy scales. Therefore, their analysis can provide information about material properties, and serve as a verification of our model, independently of the average shape analysis presented earlier.

We analyze the fluctuations in the pitch of twisted configurations in the range $10 < W < 40$ nm. The elastic energy associated with small deviation $\Delta P$ from equilibrium pitch, $P(W)$, is $\Delta E \approx Y f(W)(\Delta P)^2$. Here $Y$ is a 2D Young's modulus and $f(W)$ depends only on geometrical parameters $(t, \nu, \alpha, W, k_0)$. It is computed from our model (Supplementary notes (4) and Supplementary Fig. 10), using the same parameters as in Fig. 3d. The product $Yf(W)$ sets the standard deviation (std) of the pitch distributions, $\sigma_P(W)$, at different ribbon widths. Calculation of $\sigma_P(W)$ for twisted configurations reveals an unusual, non-monotonic dependence on $W$, indicating ribbon rigidity (which scales inversely to $\sigma_P(W)$) that first increases with $W$, but then decreases for $W > 25$ nm (Fig. 4a solid line).

We analyze ~2800 ribbons, measuring $\sigma_P(W)$ at bin size $\Delta W = 2$ nm (inset Fig. 4a). The measured $\sigma_P(W)$ is consistent with our predictions, including the predicted softening of ribbons for $W > 25$ nm (Fig. 4a). We emphasize that such unique property cannot exist in compatible twisted ribbons (as well as flat ribbons and rods) that stiffen with $W$, leading to monotonically decreasing std (Fig. 4a, dashed line). Fitting the data we find $Y \approx 9.5$ MPa consistent with measured Young's modulus of other phospholipids[30]. We group data from all widths into one distribution by rescaling $\Delta P(W)$ by the computed $Yf(W)$. Plotting the probabilities for the rescaled energy fluctuations (Fig. 4b) we find Gaussian distribution (manifested as a straight line) for moderate fluctuations (up to three std), with a systematic deviation of larger fluctuations. Plotting the normalized pitch fluctuations (inset of Fig. 4b) reveals that the deviations are all of pitch that is larger than expected by the linearized calculation, i.e., the distribution is asymmetric, having a positive skewness (>0.5). This skewness reflects a strong nonlinearity (in $\Delta P$) of the ribbon stiffness, nonlinearity which is probed only by large fluctuations. Similarly to the std, the (computed and measured) skewness is non-monotonic with $W$ (Fig. 4c and Supplementary Fig. 11), indicating increasing nonlinearity close to the twisted-to-helical transition.

## Discussion

The observed nontrivial statistics can be understood from a simplified model of incompatible purely twisted ($\alpha = 0$) ribbons:

The energy of a ribbon of thickness, $t$, width, $W$ and reference pitch, $P_0 = \frac{2\pi}{k_0}$, depends on the actual pitch, $P$, as follows:

$$E \sim \frac{tW^5}{P^4} + t^3 W \left(\frac{1}{P} - \frac{1}{P_0}\right)^2 \qquad (5)$$

The first term is the stretching energy, which (non-locally) penalizes Gaussian curvature ($K \propto \frac{1}{P^2}$). The second term is the bending energy, which penalizes deviations from the reference pitch. The combined (dimensionless) energy is plotted (Fig. 4d) as a function of $\tilde{P}$, for different $\tilde{W}$ together with an illustration of accessible states at (dimensionless) thermal energy $\widetilde{k_B T} = 0.2$ (the colored areas at each minimum). We find that as $\tilde{W}$ increases, the energy minimum shifts to larger pitch values, its depth increases up to $\tilde{W} = 0.4$ (the blue curve) but then significantly decreases, and it becomes increasingly asymmetric as $\tilde{W}$ approaches 1.

This captures the essence of shape evolution and statistical mechanics of the incompatible ribbons: due to incompatibility there is no configuration, in which the two energy terms simultaneously vanish. The stretching energy vanishes only at $\tilde{P} \to \infty$, while the bending energy vanishes at $\tilde{P} = \tilde{P}_0 = 1$. Due to the different scaling with $\tilde{W}$ of the bending ($\sim \tilde{W}$) and stretching ($\sim \tilde{W}^5$) terms, the competition between them is resolved differently, depending on $\tilde{W}$: For $\tilde{W} \ll 1$ the bending term dominates, leading to $\tilde{P} \approx \tilde{P}_0$ and a deep energy minimum. As $\tilde{W}$ increases, stretching starts dominating. As a result, the minimum is shifted to $\tilde{P} > \tilde{P}_0$ and becomes shallower and asymmetric. In addition, the total (residual) minimal energy increases with $\tilde{W}$, until the twisted solution loses stability and is replaced by the helical one (not shown). It is important to notice that the large asymmetry close to $\tilde{W} = 1$ and the resultant skewness of the distribution can lead to a significant difference between the average measured pitch and the mechanical equilibrium pitch. Such effects, which apparently are not negligible in our system might affect the estimation of the geometrical parameters from measurements and might be the source of the systematic deviation in Fig. 3d.

Finally, we note that a compatible twisted ribbon would show dramatically different statistics. In this case, the ribbon energy is of the form:
$E \sim tW^5 \left(\frac{1}{P^2} - \frac{1}{P_0^2}\right)^2 + t^3 W \left(\frac{1}{P} - \frac{1}{P_0}\right)^2$ which leads to a fixed minimum at $P = P_0$, which gets narrower and effectively more symmetric (for a fixed $T$) with $W$ (Supplementary Fig. 12), implying monotonically decreasing std and skewness. The observations in the simplified model hold for the exact calculations (Eqs. (1) and (2)).

The work shows the existence of geometrically frustrated assemblies on the sub-micron scale and presents a general way to study them: chemical information is integrated into the theory of incompatible sheets, utilizing its computing power to link between molecular and supramolecular properties of soft molecular assemblies. The quantitative modeling of chiral ribbon's shape and statistics indicates that the assemblies are incompatible ribbons with Euclidean reference metric and an asymmetric saddle reference curvature. The twisted-to-helical transition is a direct outcome of the bending-stretching competition, and very likely does not result from thermodynamic changes in the material, but only from changes in the ribbon's width. Furthermore, the non-monotonic std and skewness disclosed in this work, are unique to the modeled incompatible ribbons and cannot appear in compatible structures. We suggest that a wide range amphiphilic bilayers, as well as peptide and proteins ribbons, form frustrated ribbons of this type. The necessary conditions are ordered attractive interaction, flexibility and chirality of the building blocks. The evolution of their shape, as well as its statistics will be dominated by residual stresses. As such, they are size-dependent and should be modeled accordingly (small scale molecular simulations cannot provide the right results).

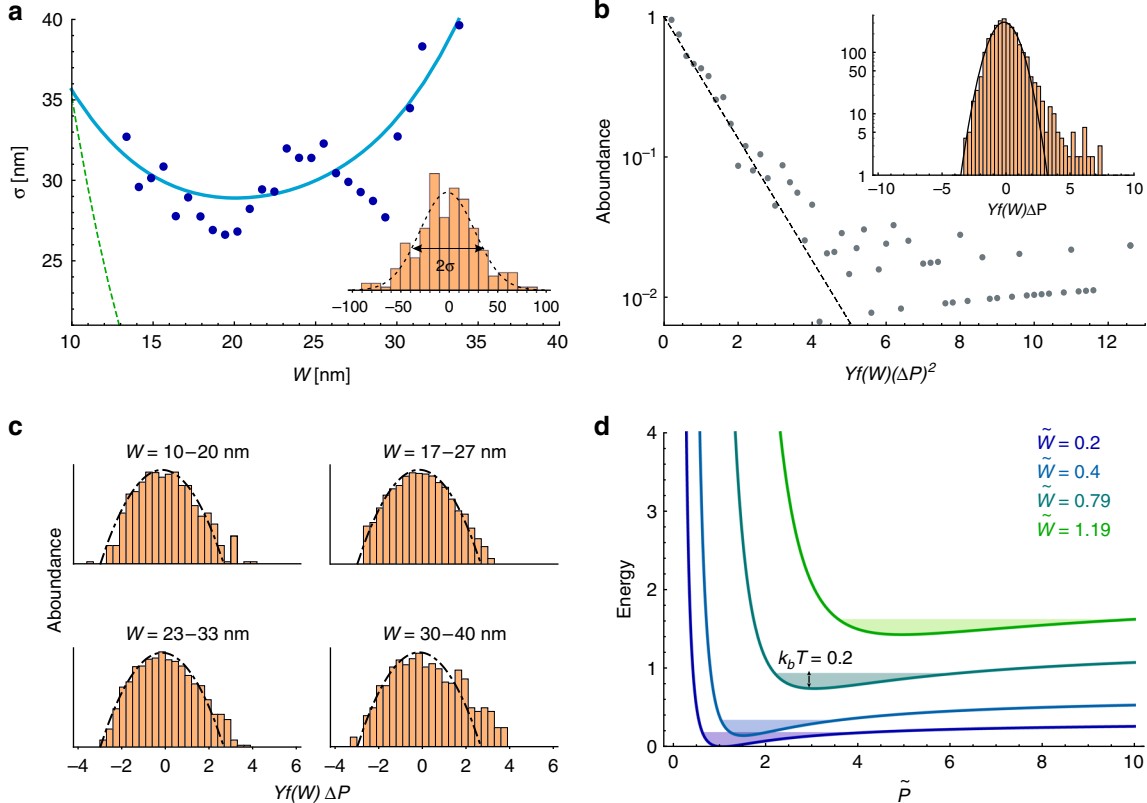

**Fig. 4** Statistics of ribbon shapes. **a** The standard deviation of the pitch of twisted ribbons as a function of ribbon width. The cyan line is the theoretical prediction, with $Y = 9.5$ *MPa*. The green dashed line is the calculated std for a compatible twisted ribbon with the same dimensions. Inset: measured pitch distribution around the average pitch and the determination of $\sigma$, obtained for $W = 21 \mp 1$ nm. **b** The abundance of ribbon configuration vs. $Yf(W)(\Delta P)^2$, (with $Yf(W)$ as in **a** and $T = 300$ $K$). An exponential dependence (dashed line) is found up to three standard deviations. Larger deviations are found to have higher probability than Gaussian. Data (~2800 data points) were collected from the entire range of widths (10−40 nm) of twisted ribbons. Inset: the distribution of normalized pitch deviations $\Delta P$ (semi log plot) with a Gaussian fit (solid line). The distribution is positively skewed. **c** Distributions of $\Delta P$ at different ranges of ribbon widths (indicated in each panel). The skewness varies non-monotonically with $W$ (with increasing width: skewness values are: 0.38, 0.22, 0.39, and 0.42). **d** The (dimensionless) energy versus pitch as calculated from Eq. (5) for twisted ribbon with $t = P_0 = 1$ for different values of $\widetilde{W}$ (indicated). The solid areas illustrate accessible states at a fixed (dimensionless) thermal energy, $\widetilde{k_B T} = 0.2$

Our nano-scale ribbons are strongly affected by thermal fluctuations and important information can be extracted from their analysis. As we showed, analysis of several cryo-TEM images provides information about the rigidity of the material (Young's modulus), nonlinearities in interaction and nearest neighbors conformations. In addition, fluctuations can qualitatively affect the coarse-grained modeling of the system. Examples are the shift of averages from mechanical equilibrium points, presented here, and the nontrivial renormalization of mechanical parameters[31]. In some cases, such effects can change a 2nd order transition to 1st order one[32,33]. Such effects possibly explain the systematic deviations in Fig. 3c, d. It is important to note that many different structures with various intrinsic geometries can be handled by combining chemical analysis with incompatible elasticity theory (see Supplementary Information (5)), i.e., by directly following the steps presented here for our specific molecule, suggesting a vast of new research, as well as application possibilities.

## Methods

**Cryo-transmission electron microscopy**. Specimens for cryo-TEM analysis were prepared in the semi-automated Vitrobot (FEI) or in the controlled environment vitrification system (CEVS), at 25 °C and water saturation to prevent evaporation from the specimens during preparation. In total, ~7 μL drop of each suspension was placed on a perforated carbon film (Ted Pella), blotted to create a thin film (manually in the CEVS or automatically in the Vitrobot), plunged into liquid

ethane (−183 °C) to create a vitrified specimen, and transferred to liquid nitrogen (−196 °C) for storage until examination. Analysis was done in the Tecnai T12 G2 TEM (FEI) at 120 kV using a Gatan 626 cryo holder maintained below −175 °C. Images were recorded on a Gatan 2kx2k UltraScan 1000 camera in the low-dose imaging mode to minimize electron-beam radiation damage[34] (See Supplementary Methods for more details).

**High-resolution scanning Electron Microscopy**. Samples were examined in a Zeiss Ultra Plus high-resolution scanning electron microscopy (HR-SEM) equipped with a Schottky field-emission electron gun at a very low electron acceleration voltage (1 kV) and short working distance (2.5–5 mm) using the Everhart-Thornley secondary electron imaging detector.

**Measurements of ribbons width and pitch**. The maturation of ribbons configuration as a function of time was analyzed by measuring ribbons' width, pitch and radius (Supplementary Fig. 2). Most ribbons have a remarkably well-defined width and pitch that vary with no more than 3% within any given ribbon (Supplementary Fig. 2). Therefore, the notions of "ribbon width" and "ribbon pitch" are well defined. The dimensions, however, vary between ribbons (Supplementary Fig. 2a).

## Data availability

The data that support the findings of this study are available from the corresponding author upon request.

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

## Acknowledgements

This research was supported by the USA-Israel binational science foundation, grant # 2014310 (ES) and the Israel Science Foundation grant No. 1117/16 (DD). M.Z. was supported by the Hebrew University Post-doctoral scholarship (PBC).

## Author contributions

M.Z. conceived the study, performed the experiments, and analyzed data. D.G. performed all the theoretical study and modeling and analyzed data, D.D. conceived the study, E.S. conceived the study. All authors co-wrote the paper.

## Additional information

**Competing interests:** The authors declare no competing interests.

**Peer Review Information:** *Nature Communications* thanks the anonymous reviewers for their contribution to the peer review of this work. Peer reviewer reports are available.

