## [Peer Review File · Nature Communications]

REVIEWERS' COMMENTS:

Reviewer #2 (Remarks to the Author):

The authors report on a new paradigm based on the theory of incompatible elasticity to predict the shape of frustrated nanoribbons obtained from the self-assembly of amphiphils via short-range interactions. The in-silico derived model is validated with high-quality cryo-TEM micrographs. This is the second time I review this paper, and the authors' response has addressed every point raised during the first round of revision in a competent manner. Therefore, I have now only minor corrections to suggest as follows:

1. Abstract. Please change “that links between molecular properties and the supramolecular structures” into “that links between molecular properties and the supramolecular morphology with its size parameters”.
2. Please proofread the MS carefully - there's a few typos
3. Fig. 2 please substitute the green colour with a primary blue or another colour that gives high contrast with the image underneath.
4. Please reword the sentence that cites refs 7-12: now it reads "peptides (7-10), as well as proteins (11,12)". This is not correct - actually refs. 8 and 11 concern an amphiphile, ref. 9 a peptide amphiphile (not simply a peptide), ref. 10 concerns Fmoc-DOPA (not a peptide, rather an amino acid derivative). Please rephrase accordingly.

Reviewer #3 (Remarks to the Author):

As before, I like this work. The authors' responses to my comments have clarified the text more. From the perspective of mechanics, it is a nice paper connecting small and larger scale principles very nicely in real systems. I think the outcome is fairly general, as long as the ribbons of interest remain well-represented within an elasticity framework. I am no chemist, though I have to imagine many nano-ribbon families would be closely represented by such a framework, since the elastic backbone seems pretty robust. I think the authors' response to reviewer 2 helps to make the connection to the broad case more clear.

We thank the reviewers for their recommendations. We implemented all the suggestions:

Referee2

1. Abstract. Please change “that links between molecular properties and the supramolecular structures” into “that links between molecular properties and the supramolecular morphology with its size parameters”.

Reply

We implemented the suggestion

2. Please proofread the MS carefully - there's a few typos

Reply

We implemented the suggestion

3. Fig. 2 please substitute the green colour with a primary blue or another colour that gives high contrast with the image underneath.

Reply

We changed the green color to orange.

4. Please reword the sentence that cites refs 7-12: now it reads "peptides (7-10), as well as proteins (11,12)". This is not correct - actually refs. 8 and 11 concern an amphiphile, ref. 9 a peptide amphiphile (not simply a peptide), ref. 10 concerns Fmoc-DOPA (not a peptide, rather an amino acid derivative). Please rephrase accordingly.

Reply

We thank the referee for the clarification. We have changed the sentence to: "... Such structures are formed by a wide variety of building blocks, such as amphiphilic lipids, peptide amphiphiles, amino acid derivatives, and proteins (1-8)."